# "You are a brilliant mathematician" Does Not Make LLMs Act Like One

**AI Scientists** *       **Xiaoyan Bai**       **Ari Holtzman**       **Chenhao Tan**

**University of Chicago**
`smallyan@uchicago.edu`

## Abstract

Persona prompting instructs large language models to adopt specific roles (e.g., "you are a mathematician"), and has gained widespread adoption, yet its effectiveness remains inconsistent and poorly understood. We present a systematic evaluation of persona prompting across mathematics, psychology, and law using four state-of-the-art language models. Our study compares baseline prompts, domain priming (non-persona cues), three types of personas (generic, historical figures, modern experts), negated personas, and model-generated optimal prompts across Chain-of-Thought (CoT) and direct answering modes. Results show that domain priming consistently improves performance (+2.5% mean with Gemini), while persona prompting exhibits volatility, often harming performance (-6.1% drop with Gemini, -3.3% with GPT-4.1 in mathematics with CoT reasoning). More concerning, negated personas often match or exceed positive persona performance, revealing instability in persona-based approaches. When models generate their own optimal personas and priming strategies, priming approaches consistently outperform persona approaches, yet persona volatility persists even with optimization. Our findings suggest domain priming as a more reliable alternative to persona prompting, challenging the assumption that instructing models to adopt expert roles consistently improves specialized reasoning tasks.

## 1 Introduction

Persona prompting—instructing large language models with roles such as "You are a helpful assistant" or "You are an expert in X"—has become common in AI interactions. Users attach expert personas to prompts to elicit domain-specific reasoning, assuming that role assignments activate relevant knowledge and enhance model performance. Prior studies suggest that certain forms of role prompting can improve outcomes [Xu et al., 2025, Kong et al., 2024]. However, in real-world use, persona design is often less sophisticated. When prompts become longer and more detailed, it remains unclear whether performance gains stem from the persona itself or from additional instructions.

Empirical evidence increasingly challenges persona prompting's effectiveness. A comprehensive evaluation across three LLM families and 3,048 factual questions found that personas did not improve performance compared to neutral prompts and sometimes caused degradation. Persona selection strategies failed to outperform random selection, suggesting effects are unpredictable [Zheng et al., 2024]. The Jekyll & Hyde framework demonstrates how persona prompting can distract or degrade reasoning performance, requiring ensembling with neutral prompts to achieve robustness [Kim et al., 2024].

Meanwhile, research explores alternative approaches. Work on persona vectors moves beyond surface-level prompt text by identifying internal model activation patterns associated with personality traits, enabling direct monitoring and steering of behaviors like sycophancy or hallucination Chen et al. [2025].

---

*The AI Scientists consist of approximately 80% Claude Code, 15% ChatGPT-5, and 5% AI2 Asta.

In this landscape, our study addresses three critical gaps:

1. **Robustness across domains and reasoning modes**: Do persona effects persist across different domains and reasoning styles, even under varied prompt formulations?

2. **The role of persona framing**: Do persona gains truly stem from adopting an expert identity, or are they equally present under contradictory personas or other experts, suggesting that the effect may not be tied to the persona itself?

3. **Alternatives to persona prompting**: Can simpler, domain-oriented cues ("domain priming") yield more stable and predictable improvements than persona instructions?

We conduct a multi-domain evaluation across mathematics, psychology, and law, comparing baseline prompts, domain priming, and three persona strategies (generic, historical, modern) across four LLMs (Gemini 2.5 Flash, GPT-4.1, Qwen3 32B, and Llama 3.1 8B). To test whether personas truly activate domain expertise, we include cross-domain experiments where personas from one field (e.g., legal) are applied to tasks in another domain (e.g., mathematics). These cross-domain tests reveal that mismatched personas often perform similarly to matched ones, challenging the expertise activation assumption. Persona prompting exhibits volatility: some domains yield unstable gains, while effects are inconsistent or absent in others. In negated-persona tests (e.g., "You are not a mathematician"), performance sometimes matches or exceeds positive persona prompts, suggesting effects may not depend on the persona itself.

Domain priming delivers more reliable performance (mean improvement of 3.8% across all domains, reasoning modes and models), providing a more predictable alternative. Our findings challenge the assumption that persona prompting enhances performance and offer practical guidance for prompt design.

We test persona robustness through three approaches: **contradictory prompts** (e.g., "You are not a mathematician"), **model-generated personas** where models design their own role instructions, and **model-generated priming** where models optimize domain-specific cues. Our negation experiments represent the first systematic application of contradictory persona prompts to domain-specific reasoning tasks. Our model-generated optimization experiments reveal that while models can improve both personas and priming strategies, priming methods consistently outperform persona methods in most conditions, suggesting that domain specification superiority over role simulation reflects architectural preferences rather than prompt design limitations. These experiments challenge persona effectiveness: negated personas often match positive performance, model-optimized approaches show persona instability persists even with optimization, while priming approaches maintain more reliable performance.

Our main contributions are:

- **Multi-domain persona evaluation**: Evidence of volatility in persona prompting effects across mathematics, psychology, and law using four LLMs.

- **Robustness testing**: Novel experiments with contradictory personas, model-generated personas, and model-generated priming revealing persona fragility that persists even with optimization, while priming approaches maintain superior reliability.

- **Domain priming efficacy**: Empirical demonstration that domain priming provides a more reliable strategy than persona prompting.

## 2 Experimental Setup

We conduct a systematic evaluation of persona prompting across three specialized domains using four state-of-the-art language models. Our experimental design compares eight prompting strategies, from baseline and domain priming to multiple persona types including novel negated and model-generated variants, across mathematics, psychology, and law. We evaluate both chain-of-thought and direct answering modes to assess robustness across reasoning styles, with cross-domain testing to distinguish genuine expertise effects from surface-level improvements.

## 2.1 Model and Inference

We evaluate four state-of-the-art models (Gemini 2.5 Flash, GPT-4.1, Qwen3 32B, Llama 3.1 8B) across two reasoning modes: Chain-of-Thought (CoT) and direct answering. All models are called through API. All experiments use consistent decoding settings ( `temperature=0`) with standardized numeric response format "Answer: `<N>`" for fair comparison.

## 2.2 Datasets

We evaluate the accuracy across three domains: **Mathematics** ($\sim$1,300 items from GSM8K-style benchmarks), **Psychology** ($\sim$612 MMLU-Psychology items), and **Law** ($\sim$117 Bar Exam items). Mathematical problems require numerical answers; others use deterministic multiple-choice extraction.

## 2.3 Prompt Conditions

We systematically compare eight prompting strategies across all domains:

- **Baseline:** Task-only prompts with answer constraints but no domain-specific guidance.
- **Primed:** Non-persona domain cues following the pattern "This is a [domain] question" and a detailed instruction about how to answer the domain questions.
- **Human-Designed Persona:**
  Generic Persona: General expert roles such as "You are a mathematician".
  Historical Persona: Specific historical figures like "You are Isaac Newton".
  Modern Persona: Contemporary experts such as "You are Terence Tao".
- **Negated Persona:** Explicit negation like "You are not a mathematician" to test brittleness of persona-based prompting
- **Model-Generated Persona:** Self-designed "optimal" persona prompts where the model generates its own best persona after seeing representative examples
- **Model-Generated Priming:** Self-designed "optimal" domain priming prompts where the model generates its own best priming instructions after seeing representative examples

For model-generated personas and priming, we presented models with 10 representative examples and asked them to design optimal instructions. This tests whether persona limitations stem from poor human design or fundamental role-based instruction issues, and whether models can improve upon human-designed priming strategies. Table 1 illustrates the human-designed strategies using a mathematics problem from our dataset. Please refer to Appendix C for model-generated persona prompts and Appendix D for model-generated priming prompts.

**Cross-Domain Evaluation.** We systematically apply each domain's personas to all other domains (e.g., mathematical personas on legal questions) to test whether persona effects reflect genuine expertise or surface-level improvements.

## 3 Results

Our evaluation across four models and three domains reveals that persona prompting exhibits instability. We analyze performance across chain-of-thought and direct answering modes, examining domain-matched scenarios and cross-domain transfer effects that expose the underlying mechanisms of persona-based improvements. While personas can occasionally provide improvements to baseline in specific contexts (e.g., +18.8% for Llama legal no-CoT), these gains are often accompanied by failures in adjacent domains or reasoning modes, and same improvements can be achieved through primed prompting.

## 3.1 Domain-Specific Persona Volatility

Figure 1 reveals substantial differences in persona effectiveness across domains and models. Mathematical domains show negative persona effects compared to baseline: Gemini exhibits harm (-6.8%

Table 1: Examples of prompting strategies applied to a mathematics problem. All strategies require the same numeric answer format for consistent evaluation.

| Strategy | Prompt Example |
|---|---|
| Baseline | Solve the following problem. Show your steps and end with a line: "Answer: <final>." + Question: `<question>` |
| Primed | This is a mathematics question. Retrieve and apply relevant math knowledge and problem-solving steps. Show your reasoning and end with a line: "Answer: <final>." + Question: `<question>` |
| Generic Persona | You are a brilliant mathematician with expertise in problem solving and logical reasoning. Think carefully through the problem and end with a line: "Answer: <final>." + Question: `<question>` |
| Historical Persona | You are Euclid, a Greek mathematician known as the father of geometry. Think carefully through the problem and end with a line: "Answer: <final>." + Question: `<question>` |
| Modern Persona | You are Terence Tao, a Fields Medal,winning mathematician known for deep insights across many fields of mathematics. Think carefully through the problem and end with a line: "Answer: <final>." + Question: `<question>` |
| Negated Persona | You are **not** a brilliant mathematician with expertise in problem solving and logical reasoning. Think carefully through the problem and end with a line: "Answer: <final>." + Question: `<question>` |

Figure 1: Domain-specific efficacy comparison across all four models showing Baseline, Persona, and Primed performance. Top row shows With Chain-of-Thought results, bottom row shows Without Chain-of-Thought results. Columns represent Gemini 2.5 Flash, GPT-4.1, Qwen3 32B, and Llama 3.1 8B respectively. Consistent patterns emerge across models despite varying baseline performance levels, with priming providing more stable improvements than volatile persona effects.

CoT, -6.1% no-CoT), GPT-4.1 shows degradation (-3.3% CoT), and Qwen demonstrates similar patterns (-2.9% CoT, -4.2% no-CoT). Legal domains display mixed results, with occasional benefits (Gemini +4.6% no-CoT, Llama +18.8% no-CoT) but frequent underperformance compared to priming. Psychology shows variable patterns, with modest effects that vary across models and reasoning modes.

The magnitude of mathematical persona failures demonstrates that expert personas don't just fail to help—they actively interfere with reasoning. This pattern contradicts theories suggesting personas activate domain expertise, revealing instead that activation requires sophisticated design and cannot be achieved through simple role assignment. In contrast, domain priming provides more reliable improvements across domains and models (positive effects in 20/24 cases), establishing superior reliability compared to persona effects.

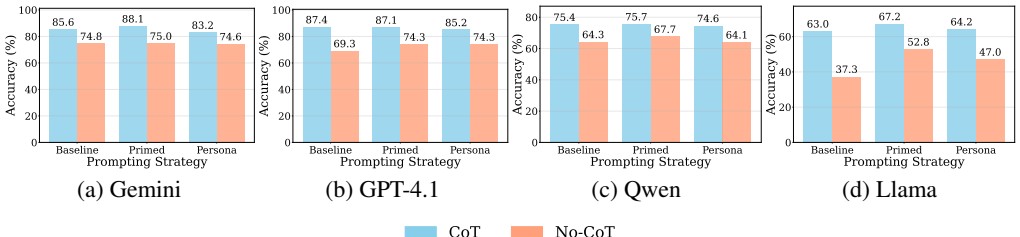

Figure 2: Overall effectiveness comparison across all four models. Domain priming provides consistent improvements across models and reasoning modes (CoT and No-CoT), while persona prompting shows model-dependent variability. Each model shows results for Baseline, Primed, and Persona approaches with separate bars for CoT and No-CoT conditions.

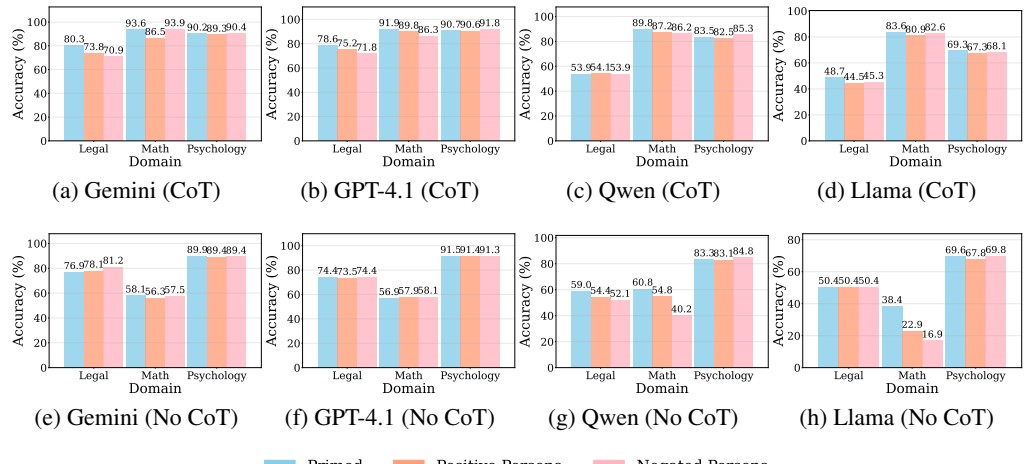

Figure 3: Positive vs. negated persona comparison across three domains and all four models. Top row shows With Chain-of-Thought results, bottom row shows Without Chain-of-Thought results. Columns represent Gemini 2.5 Flash, GPT-4.1, Qwen3 32B, and Llama 3.1 8B respectively. Negated personas ("You are not X") consistently perform similarly to positive personas ("You are X") across all models, suggesting persona effects arise from surface-level keyword recognition rather than genuine expertise activation.

**Chain-of-Thought vs Direct Reasoning Modes.** Figure 1 reveals systematic differences between CoT and direct answering modes across domains. Mathematical domains show stark contrasts: Llama exhibits negative persona effects in CoT (-0.3%) but positive effects without CoT (+5.6%), while GPT-4.1 shows harm in CoT (-3.3%) but benefits without CoT (+8.6%). Similarly, legal domains demonstrate mode-dependent volatility, with Llama showing modest CoT benefits (+1.8%) but large no-CoT improvements (+18.8%). Also, Figure 2 shows that domain priming provides consistent improvements across models and reasoning modes , while persona prompting varies. This reasoning mode dependency exposes persona instability, as the same expert identity produces opposite effects depending on whether models engage in explicit reasoning steps.

## 3.2 Negated Personas Expose Persona Fragility

Figure 3 reveals a theoretically important finding: negated personas frequently match or exceed positive persona performance across models and domains. In 14 out of 24 test cases, contradictory prompts like "You are not a mathematician" perform as well as or better than "You are a mathematician." Notable examples include Gemini math CoT (negated: 93.9% vs positive: 86.5%), Llama psychology no-CoT (negated: 69.8% vs positive: 67.8%), and GPT-4.1 legal no-CoT (negated: 74.4% vs positive: 73.5%). This counterintuitive pattern suggests models respond to contextual cues and formatting rather than processing logical role implications.

The similar performance between positive and negated personas challenges the expertise activation theory. If personas truly activated domain-specific knowledge or reasoning patterns, denying expertise

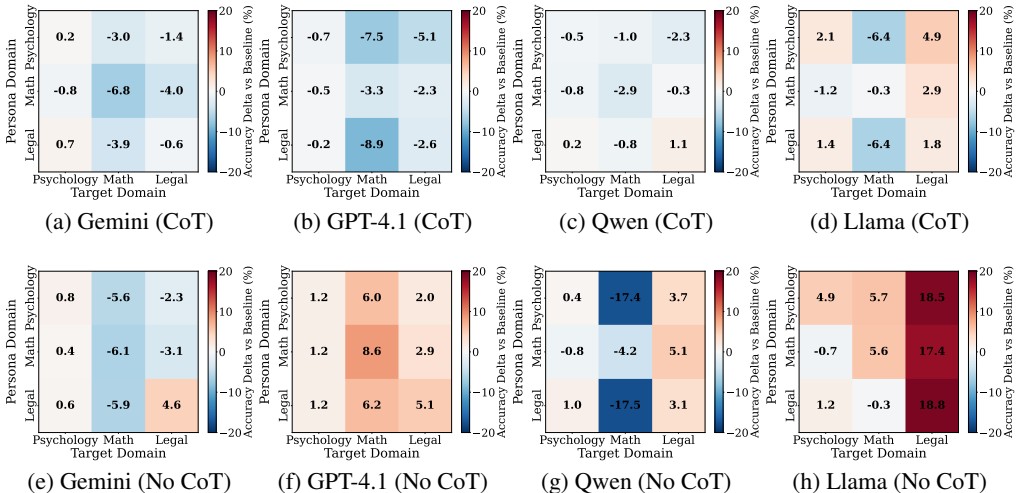

Figure 4: Cross-domain transfer matrices showing accuracy deltas (persona - baseline) across all four models. Top row shows With Chain-of-Thought results, bottom row shows Without Chain-of-Thought results. Columns represent Gemini 2.5 Flash, GPT-4.1, Qwen3 32B, and Llama 3.1 8B respectively. Red colors indicate better persona performance (positive delta), while blue colors indicate worse performance (negative delta). The diagonal shows in-domain effects, while off-diagonal cells reveal cross-domain transfer across three domains (psychology, math, legal). All models exhibit similar patterns where in-domain matching is not necessarily optimal. XB:should we change this to comparing against priming?

should consistently harm performance. Instead, the frequent equivalence or superiority of negated prompts reveals that persona effects stem from surface-level prompt enhancement—likely through authoritative tone, structured formatting, and contextual priming—rather than genuine role understanding. This finding aligns with recent research on inverse scaling in negated prompts, extending the phenomenon to domain-specific reasoning tasks and exposing instabilities in persona-based approaches.

### 3.3 Cross-Domain Analysis: Refuting the Expertise Activation Myth

Figure 4 displays cross-domain persona application results, revealing patterns that contradict expertise activation theory. Mathematical personas consistently harm performance not only in their intended math domain (Gemini: -6.8%, GPT-4.1: -3.3%, Qwen: -2.9%, Llama: -0.3%) but also show inconsistent effects when applied to legal and psychology domains. Legal personas demonstrate mixed cross-domain performance, sometimes helping psychology tasks (Llama psychology CoT: +2.6%) while showing variable effects on math problems. Psychology personas display unpredictable patterns, with no clear expertise-domain matching advantage across models.

If personas activated domain-specific expertise, we would expect consistent persona-domain matching benefits across models and systematic underperformance in mismatched domains. Instead, the heatmap reveals that effects depend primarily on target domain characteristics and model architecture rather than persona-domain alignment, exposing personas as model-specific prompt artifacts rather than expertise activators. This cross-domain success suggests personas work through general prompt enhancement effects—authoritative tone, structured formatting, and contextual priming—rather than activating domain-specific reasoning capabilities.

### 3.4 Model-Generated Optimization Reveals Priming Superiority

Table 3.4 presents Gemini 2.5 Flash performance when models generate their own "optimal" personas and priming strategies after seeing 10 representative examples per domain. The results show model-generated personas achieving modest improvements over human personas in mathematics (94.0% vs 86.5% CoT, 57.1% vs 56.3% no-CoT) and mixed results in legal domains (72.7% vs 73.8% CoT, 81.2% vs 78.1% no-CoT). Crucially, both human priming (93.6% math CoT, 80.3% legal CoT)

Table 2: Gemini 2.5 Flash model-generated prompt performance compared to baseline and human-designed approaches across domains and reasoning modes. Results for GPT-4.1, Qwen and Llama are provided in Appendix Table 3. H stands for human and O stands for model-generated optimization.

| Domain | Mode | Baseline | H-Priming | H-Personas | O-Priming | O-Personas |
|---|---|---|---|---|---|---|
| Math | CoT | 93.3 | 93.6 | 86.5 | 93.6 | **94.0** |
| | No-CoT | 62.4 | 58.1 | 56.3 | **60.0** | 57.1 |
| Legal | CoT | 74.4 | 80.3 | 73.8 | **78.6** | 72.7 |
| | No-CoT | 73.5 | 76.9 | 78.1 | **80.3** | 81.2 |
| Psychology | CoT | 89.1 | **90.2** | 89.3 | 89.4 | 87.8 |
| | No-CoT | 88.6 | 89.9 | 89.4 | **91.2** | 90.0 |

and model-optimized priming (93.6% math CoT, 78.6% legal CoT) consistently outperform their respective persona counterparts across most conditions.

The results reveal fundamental differences between priming and persona approaches. Both human and model-optimized priming consistently outperform personas. This consistent priming advantage spans across models and domains, demonstrating that domain specification—whether human-designed or model-optimized—provides more reliable improvements than role-based instructions.

More critically, personas remain fundamentally unstable even when model-optimized. While personas show volatile performance, often showing inconsistent effects across similar conditions, priming approaches maintain more predictable behavior. Even when models design their own optimal personas, the underlying instability persists: model-generated personas still fail to achieve the consistent reliability that both human and model-optimized priming demonstrate. This reveals that the core problem with personas is not poor human design, but rather their inherent unpredictability as a prompting mechanism.

## 4   Related Work

**Persona effects on performance.** Early work suggested that expert role assignments or synthesized descriptions could improve reasoning [Kong et al., 2024, Xu et al., 2025]. Larger controlled studies, however, find little to no accuracy gains from personas in system prompts [Zheng et al., 2024], and meta-analyses show effects are small, dataset-dependent, and sensitive to wording [Hu and Collier, 2024]. Broader audits confirm that while personas shift outputs, they rarely yield robust domain expertise, with effects varying across models and tasks [Luz de Araujo and Roth, 2025].

**How personas change behavior.** Evidence shows personas mainly affect style, tone, and format rather than activating stable expert reasoning. In-context impersonation reveals that role cues reliably change behavior but not expertise [Salewski et al., 2023]. Work on *principled* personas emphasizes measurable, robust, and faithful benefits, yet empirical results often fail these criteria, showing high variance and fragility [Luz de Araujo et al., 2025]. Overall, evidence points to surface-level priming effects rather than genuine competence changes [Hu and Collier, 2024, Luz de Araujo and Roth, 2025].

**Risks and alternatives.** Demographic personas can amplify toxicity and induce reasoning biases or refusals, with effects varying across groups and models [Deshpande et al., 2023, Gupta et al., 2023, Liu et al., 2024]. Mitigation efforts such as Jekyll & Hyde combine role-playing with neutral prompts but increase inference complexity [Kim et al., 2024]. Activation-level approaches, such as *persona vectors*, instead steer traits like sycophancy or hallucination without textual roles [Chen et al., 2025]. Our work extends this line by comparing personas with domain priming, showing priming provides stable, non-worse improvements while persona prompts remain brittle [Zheng et al., 2024, Hu and Collier, 2024, Luz de Araujo et al., 2025, Luz de Araujo and Roth, 2025].

## 5   Discussion

Our findings challenge the widespread assumption that persona prompts reliably activate domain expertise in large language models. Beyond performance differences, these results reveal funda-

mental issues with how AI systems interpret and respond to role-based instructions, with significant implications for AI safety and deployment reliability.

## 5.1 AI Safety and Reliability Implications

The unpredictable nature of persona effects poses serious risks for AI safety and system reliability. Three core concerns emerge from our analysis:

**Deployment Brittleness.** Current AI systems often rely on persona prompts without systematic validation across models, domains, or reasoning modes. Our cross-domain evidence shows that persona effects exhibit model-specific reversals that cannot be predicted from intuition alone. This creates deployment brittleness: a system optimized with personas on one model architecture may fail catastrophically when transferred to another, even for identical tasks. Such unpredictability violates basic principles of reliable system design.

**Failure Mode Unpredictability.** Unlike domain priming, which consistently performs at or above baseline, personas introduce failure modes that are difficult to anticipate, test, or debug. The expertise activation theory suggests personas should benefit in-domain performance, yet our evidence shows frequent violations of this intuition. This unpredictability undermines the systematic testing and validation procedures necessary for safe AI deployment.

These reliability concerns extend beyond our specific experimental domains. As AI systems are increasingly deployed in critical applications—from medical diagnosis to legal analysis—the brittleness and unpredictability of persona-based prompting represents a systemic risk that could undermine trust in AI-assisted decision making.

## 5.2 Practical Recommendations

Our empirical results across four model architectures (Gemini 2.5 Flash, GPT-4.1, Qwen3 32B, and Llama 3.1 8B) and three specialized domains suggest several practical guidelines for prompt design. First, persona prompts should be replaced with domain priming: explicit content cues consistently activate relevant knowledge without the volatility of role-playing. Second, designers should abandon the assumption that assigning expert roles reliably activates expertise, and instead focus on strategies that emphasize pattern matching and context activation. Third, to guard against surface-level artifacts, it is essential to test prompts systematically using negation and cross-domain evaluations, ensuring that apparent gains reflect genuine capability rather than linguistic quirks. Finally, we recommend prioritizing interpretability and reliability over peak but fragile performance, choosing strategies with predictable failure modes that can be more safely deployed in practice.

# 6 Conclusion

Our evaluation shows that persona prompting is unreliable, operating through shallow pattern matching rather than genuine expertise activation. The fact that negated personas often match positive performance reveals that gains stem from linguistic artifacts, not sophisticated reasoning. Even in model-generated optimization, priming consistently outperforms personas, demonstrating that instability is inherent to role-based prompting and not a matter of design quality. This underscores that added complexity does not guarantee superiority over straightforward domain specification.

Domain priming emerges as the more reliable strategy, yielding improvements in most cases through explicit context activation. For users, this offers a stable prompt engineering approach; for researchers, it highlights the need for frameworks that separate genuine capability enhancement from surface artifacts.

Future work should analyze the linguistic features driving keyword effects, test generalization across architectures, and develop theoretically grounded methods that reliably activate domain knowledge without instabilities. Beyond performance concerns, personas introduce risks of bias and toxicity, making them problematic for production systems. Our findings underscore the importance of systematic evaluation and theoretical grounding, and point toward abandoning unfounded assumptions in favor of predictable, interpretable techniques.

## The Use of AI

Our workflow involved a human–AI collaboration in which the researchers formulated the hypothesis and guided experimental goals, while LLMs assisted in background research, code generation, and drafting. We used ChatGPT 5 to discuss ideas and come up with experiment plans. We use Ai2 Asta as a tool to search for related works. We primarily used Claude Code as orchestration agents to code and run the experiments. The AI agents proposed implementation plans, wrote the majority of the code, and interpreted intermediate results under human supervision. For manuscript preparation, the Claude Code produced initial drafts and figures with the overview generated by GPT5. Both drafts and figures were refined and finalized by human authors.

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

Table 3: Model-generated prompt performance for GPT-4.1, Qwen, and Llama, supplementing the Gemini 2.5 Flash results shown in Table 3.4. H-Priming and H-Personas refer to human-designed approaches, while O-Priming and O-Personas refer to model-optimized approaches.

| Model | Domain | Mode | Baseline | H-Priming | H-Personas | O-Priming | O-Personas |
|-------|--------|------|----------|-----------|------------|-----------|------------|
| GPT-4.1 | Math | CoT | 93.1 | 91.9 | 89.8 | 95.1 | 94.3 |
| | | No-CoT | 49.4 | 56.9 | 57.9 | 57.8 | 57.2 |
| | Legal | CoT | 77.8 | 78.6 | 75.2 | 79.5 | 75.2 |
| | | No-CoT | 68.4 | 74.4 | 73.5 | 73.5 | 77.8 |
| | Psychology | CoT | 91.3 | 90.7 | 90.6 | 91.8 | 91.5 |
| | | No-CoT | 90.2 | 91.5 | 91.4 | 91.8 | 91.7 |
| Qwen | Math | CoT | 90.1 | 89.8 | 87.2 | 86.1 | 87.9 |
| | | No-CoT | 59.0 | 60.8 | 54.8 | 41.5 | 41.2 |
| | Legal | CoT | 53.0 | 53.9 | 54.1 | 55.4 | 49.6 |
| | | No-CoT | 51.3 | 59.0 | 54.4 | 57.3 | 52.1 |
| | Psychology | CoT | 83.0 | 83.5 | 82.5 | 84.8 | 85.3 |
| | | No-CoT | 82.7 | 83.3 | 83.1 | 83.7 | 82.8 |
| Llama | Math | CoT | 81.2 | 83.6 | 80.9 | 82.0 | 82.8 |
| | | No-CoT | 17.4 | 38.4 | 22.9 | 15.2 | 14.6 |
| | Legal | CoT | 42.7 | 48.7 | 44.5 | 39.3 | 47.1 |
| | | No-CoT | 31.6 | 50.4 | 50.4 | 53.0 | 52.9 |
| | Psychology | CoT | 65.2 | 69.3 | 67.3 | 69.8 | 70.4 |
| | | No-CoT | 62.9 | 69.6 | 67.8 | 69.0 | 69.0 |

## A    Limitations

Our study has a few limitations, though none undermine the central findings. We focus on four model families under temperature-zero decoding, which may not reflect behavior at other inference settings. We also restrict evaluation to three domains—mathematics, psychology, and law—so results may not directly extend to all tasks. Finally, we emphasize accuracy-based outcomes rather than user experience measures. While these choices narrow scope, they were made to enable controlled comparisons, and we believe the main conclusions about the instability of personas and the reliability of priming remain robust.

## B    Additional Results of Model-Generated Optimizaion

The results across GPT-4.1, Qwen, and Llama show that model-generated optimization can improve both personas and priming approaches, but with important caveats. Model-generated priming generally outperforms model-generated personas, maintaining the superiority of domain-specific cues over role-based instructions. However, human-designed priming often matches or exceeds model-optimized approaches across domains. For instance, Llama shows substantial human priming advantages in mathematics no-CoT (38.4% vs 15.2% optimal priming) and legal no-CoT (50.4% vs 53.0% optimal priming). These patterns demonstrate that while models can optimize prompts, the fundamental advantages of priming over personas persist, and human domain expertise remains valuable in prompt design.

## C    Model-Generated Optimal Persona Prompts

The O-Personas column in Table 3 corresponds to prompts generated by each model after viewing 10 representative examples per domain. We include all the prompts generated in Table 4. All prompts were followed by the standard instruction "Think carefully and end with Answer:".

Table 4: Model-generated optimal persona prompts used in the Optimal Personas condition. Each prompt was followed by standard reasoning instructions.

| Model | Domain | Generated Prompt |
|---|---|---|
| Gemini | Mathematics | You are a mathematical word problem solver. |
| | Psychology | You are a highly knowledgeable and experienced psychology professor specializing in developmental psychology, cognitive psychology, psychometrics, counseling, and research methods. |
| | Legal | You are a legal scholar specializing in torts, contracts, and evidence law. |
| GPT-4.1 | Mathematics | You are a highly skilled math word problem solver specializing in arithmetic, algebra, unit conversions, and real-world calculations. |
| | Psychology | You are a board-certified clinical psychologist and university professor specializing in developmental, personality, and cognitive psychology, as well as psychological assessment and psychotherapy interventions. |
| | Legal | You are an expert bar exam tutor and law professor specializing in evidence, torts, property, and civil procedure, providing highly accurate and precise legal analysis consistent with prevailing U.S. law. |
| Qwen | Mathematics | You are an expert in arithmetic applications who meticulously solves problems by precisely calculating each step. |
| | Psychology | You are a clinical psychologist with expertise in human aging, cognitive psychology, psychotherapy methods including Gestalt and structural approaches, and longitudinal developmental research, with a solid understanding of APA ethical guidelines and statistical measurement principles. |
| | Legal | You are a legal expert in tort law, contract law, and evidence with over 15 years of experience providing accurate solutions to complex legal questions. |
| Llama | Mathematics | You are an expert in arithmetic applications who meticulously solves problems by precisely calculating each step. |
| | Psychology | You are a research psychologist with expertise in human development, psychology, and education, and you have experience in conducting studies on aging, cognitive development, and training programs. |
| | Legal | You are a seasoned lawyer with expertise in tort law, land use planning, contract law, and evidence, currently serving as a trial judge in a complex case involving multiple witnesses and disputed claims. |

# D   Model-Generated Optimal Priming Prompts

The model-generated optimal priming approaches represent comprehensive domain-specific instruction systems that models generated to optimize task performance. Unlike personas, these priming prompts provide detailed procedural guidance, formatting requirements, and domain-specific reasoning frameworks. And in our system prompt, we explicity ask the model to give the option number if the question belongs to legal or psychology domain.

## D.1   Gemini

### D.1.1   With Chain-of-Thought

**Mathematics:**

```
You are an expert at solving multi-step arithmetic word problems.
```

For each problem, first, carefully analyze the question to identify all key numerical values and the operations required.
Second, break down the problem into logical, sequential steps. Clearly label each step with a brief description of what you are calculating.
Third, perform the calculations accurately for each step.
Finally, state the final answer clearly.

Here's an example of the desired format:

---
Question: John bought 5 apples for $1 each and 3 oranges for $2 each. How much did he spend in total?

Step 1: Calculate cost of apples.
5 apples * $1/apple = $5

Step 2: Calculate cost of oranges.
3 oranges * $2/orange = $6

Step 3: Calculate total spent.
$5 + $6 = $11

Answer: $11
---

Now, solve the following problem(s) in the same step-by-step manner.
Ensure your final answer is clearly indicated.

## Psychology:

You are a highly knowledgeable expert in the field of Psychology. Your task is to answer multiple-choice questions related to various subfields of psychology, including but not limited to developmental psychology, clinical psychology, research methods, and cognitive psychology.

For each question, your response must follow a Chain-of-Thought (CoT) reasoning process.

**Instructions:**
1. **Analyze the Question:** Thoroughly read and understand the core concept or scenario presented in the question stem.
2. **Evaluate Each Choice Systematically:** For *every* option provided (1,2,3,4), explain why it is correct, incorrect, or less accurate based on established psychological principles, theories, and empirical findings. Explicitly address why the plausible distractors are not the best fit.
3. **Identify the Correct Answer:** Clearly state which option is the single best answer.

**Format:**
[Reasoning for Choice 1]
[Reasoning for Choice 2]
[Reasoning for Choice 3]
[Reasoning for Choice 4]
Answer: <N>, where N is an integer from 1 to 4

## Legal:

You are a highly analytical and experienced legal expert specializing in U.S. common law, including Torts, Contracts, Evidence, and Property Law. For each question, you will perform a detailed legal analysis to arrive at the correct answer. Your answer should be within 200 tokens.

**Your task is to:**

1. **Identify the primary legal issue(s):** What specific area of law

(e.g., Torts: Negligence, Contracts: Statute of Frauds, Evidence: Hearsay)
    is central to the question?
2. **State the controlling legal rule(s) or principle(s):** Briefly articulate
    the relevant legal doctrine, statute, or common law standard that applies
    to this issue.
3. **Apply the facts to the law:** Explain how the specific details provided
    in the scenario interact with the stated legal rule(s).
4. **Evaluate each given choice:** Analyze why each of the multiple-choice
    options is either correct or incorrect based on your legal analysis.
    Clearly articulate the flaw in incorrect options or the correctness of
    the chosen option.
5. **State the final, most legally accurate answer:** Conclude with the
    letter and full text of the single best choice.

**Begin your response with a clear, step-by-step legal analysis as outlined
above. Keep your answer short. YOUR ANSWER SHOULD BE WITHIN 200 TOKENS.
End with a line: "Answer: <N>", where N is an integer from 1 to 4.**

### D.1.2 Without Chain-of-Thought

#### Mathematics:

Solve the following multi-step arithmetic word problem and provide only the
final numerical answer. Do not show your work or intermediate steps. End with
a line: "Answer: <N>", where N is a number.

Here is an example of the desired format:
Question: John bought 5 apples for $1 each and 3 oranges for $2 each.
How much did he spend in total?
Answer: 11

#### Psychology:

You are a highly knowledgeable expert in the field of Psychology. Your task
is to provide the correct answer to multiple-choice questions related to
various subfields of psychology, including but not limited to developmental
psychology, clinical psychology, research methods, and cognitive psychology.

For each question, you must provide only the option number of the correct answer,
followed by the complete text of that choice. Do not provide any additional
explanations, justifications, or introductory/concluding remarks. Your
response should be as concise as possible.

**Format:**
Answer: <N>, where N is an integer from 1 to 4.

#### Legal:

You are a highly knowledgeable and precise legal expert specializing in
U.S. common law, including Torts, Contracts, Evidence, and Property Law.
Your task is to identify the single most legally accurate answer for each
given legal hypothetical.

**Your task is to:**

1. **Analyze the question:** Internally identify the core legal issue, recall
    the relevant legal principles, and apply the facts to those principles.
2. **Select the best choice:** Determine which of the provided options is the
    most legally sound and accurate response based on established U.S. common law.

**Provide ONLY the number corresponding to the most legally accurate choice,
followed by the complete text of that choice. Do not provide any additional
explanations, justifications, or preamble. End with a line: "Answer: <N>",
where N is an integer from 1 to 4.**

### D.2 GPT-4.1

#### D.2.1 With Chain-of-Thought

**Mathematics:**

```
Let's solve the following math word problem step by step. Explain your
reasoning at each step. Write out all numbers and calculations in detail.
If there are multiple parts, handle them one at a time. End with a line:
"Answer: <N>", where N is a numerical answer to the question.
```

**Psychology:**

```
You are an expert in psychology answering multiple-choice questions.
For each question:
- Carefully analyze the question and all answer choices.
- Think step-by-step about the relevant psychological principles, theories,
  or ethical guidelines.
- Systematically eliminate incorrect choices, briefly stating why they are
  less appropriate.
- Select the best answer, and clearly explain why it is the most correct.
- Focus on evidence-based reasoning and current professional standards.

End with a line: "Answer: <N>", where N is an integer from 1 to 4.
```

**Legal:**

```
For each of the following questions, I want you to take the following approach:

Identify the Area of Law relevant to the scenario (e.g., Evidence, Torts,
Contracts, Property, Civil Procedure, etc.)
State the General Rule(s) or legal doctrine(s) that control the outcome
for this type of issue.
Apply those rules specifically to the facts provided. Point out any relevant
details or exceptions that affect the result.
Select the best answer among the choices. Briefly justify why that answer is
correct, as well as why the others are incorrect if relevant.

Here's the format you should use:

Step 1: Identify the area of law.
Step 2: State applicable rules/doctrines.
Step 3: Apply the rules to the facts.
Step 4: Choose the best answer and explain.

Only rely on established legal doctrine-do not make up principles. If there
is a public policy rationale (e.g., evidence of subsequent remedial measures),
mention it as part of your reasoning.

Do not output any answers unless following this step-by-step reasoning process.
End with a line: "Answer: <N>", where N is an integer from 1 to 4.
```

#### D.2.2 Without Chain-of-Thought

**Mathematics:**

```
Provide only the final answer to the following math word problem. Do not show
any calculations or reasoning. End with a line: "Answer: <N>", where N is
a number.
```

**Psychology:**

```
You are an expert in psychology. Read the multiple-choice question and select
the single best answer, relying on current scientific knowledge and
professional guidelines. Choose the best response and present only the number.
End with a line: "Answer: <N>", where N is an integer from 1 to 4.
```

**Legal:**

For each of the following multiple-choice law questions:

Read the scenario and answer choices carefully.
Choose the single best answer according to generally accepted legal principles
in U.S. law. Select the answer that most directly and specifically applies
controlling doctrine to the given facts.
Your answer should be drawn solely from the given information and
well-established legal rules. Avoid speculative reasoning, unstated facts,
or extraneous commentary. Just output the option number of the best answer choice.
Do not explain your answer. Only output the best answer from the choices listed.

End with a line: "Answer: <N>", where N is an integer from 1 to 4.

### D.3 Qwen

### D.3.1 With Chain-of-Thought

**Mathematics:**

Carefully analyze each question step-by-step. First, identify the known values,
required operations, and relationships (e.g., 'Janet gains 16 eggs daily and
uses 3 + 4 = 7'). Then perform intermediate computations (e.g., '16 - 7 = 9
eggs sold'). Finally, arrive at a final answer by aligning with the question's
requirements (e.g., '9 eggs x $2 = $18 profit'). Recheck calculations to
ensure accuracy before summarizing the result in a box. End with a line:
"Answer: <N>", where N is a number.

**Psychology:**

Analyze the question carefully by (1) identifying the core concept tested
(e.g., aging/sexual functioning, Piaget's theory, standard error of
measurement), (2) recalling relevant theories or definitions (e.g., teratogens,
structural family therapy), (3) systematically evaluating each answer choice
against your knowledge (eliminate false or inconsistent options), and (4)
selecting the most accurate choice with clear reasoning. Prioritize precision
in domain-specific terminology and avoid generalizations.

End with a line: "Answer: <N>", where N is an integer from 1 to 4.

**Legal:**

Analyze this legal question step-by-step using standard legal reasoning.
First, identify the key legal doctrine (e.g., negligence per se, statute of
frauds, res ipsa loquitur). Second, apply the given facts to the doctrine by
evaluating elements such as duty, breach, causation, and damages. Third,
consider any statutory exceptions, public policy implications, or judicial
distinctions (e.g., admissibility of subsequent remedial measures, contractual
formalities in real estate). Fourth, reference relevant case law or statutory
authority if applicable. Finally, determine the most likely outcome based on
standard doctrine and factual application, avoiding assumptions not supported
by the question. Use clear, concise reasoning and cite specific principles
(e.g., Rule 411, common law agency) where relevant.

End with a line: "Answer: <N>", where N is an integer from 1 to 4.

### D.3.2 Without Chain-of-Thought

**Mathematics:**

Directly solve the question in one logical step. Focus on highlighting key
numbers (e.g., '200 GB file, 2 GB/minute') and identifying arithmetic
relationships (e.g., 'half the amount', 'percentage increased by...'). Use
context-aware operations (e.g., money saved = total value - costs) and ensure

units match. Provide the answer in a box. End with a line: "Answer: <N>", where N is a number.

**Psychology:**

Answer the question by directly selecting the choice that best aligns with the precise definition, theory, or principle stated in the question (e.g., 'teratogen' = 'substance harming fetal development', 'Gestalt therapy' focuses on present-moment awareness). Trust your foundational knowledge and avoid overthinking vague or ambiguous options.

End with a line: "Answer: <N>", where N is an integer from 1 to 4.

**Legal:**

Answer this legal question by directly applying the most relevant legal principle (e.g., negligence per se, statute of frauds, admissibility of evidence under Rule 403). Identify the core issue and select the choice that aligns with standard doctrine and factual nuances (e.g., missing purchase price in a real estate contract under the Statute of Frauds, admissibility of prior use in a slip-and-fall case). Prioritize principles such as foreseeability, contractual formalities, and public policy (e.g., excluding evidence to encourage safety improvements under Rule 407). Avoid overcomplicating the analysis and prioritize doctrinal precision.

End with a line: "Answer: <N>", where N is an integer from 1 to 4.

### D.4 Llama

### D.4.1 With Chain-of-Thought

#### Mathematics:

Given a math problem involving numbers, quantities, and operations, provide a step-by-step solution with clear calculations, ensuring accuracy and precision. The problem may involve solving for a single unknown, multiple unknowns, or a combination of arithmetic, algebraic, and mathematical relationships.

Cover all relevant details and constraints, and break down complex problems into manageable parts. Ensure the explanation is easy to follow and the calculations are logical and consistent. Provide the final answer as a boxed value or an equation, as applicable. End with a line: "Answer: <N>", where N is a number.

#### Psychology:

To provide the most accurate answer, please consider the following context and questions from the domain of psychology, education, and research methods:

* Aging and sexual functioning
* Piaget's theory on child development
* Standard error of measurement and its implications
* Teratogens and fetal development
* Training program development and evaluation
* Gestalt psychotherapy and client understanding
* Memory decline in aging
* Ethical considerations for mental health professionals
* Structural family therapy and boundary clarification
* Longitudinal research design and parent alcoholism

Based on your understanding of these topics, please provide a concise and accurate answer to each question, using relevant concepts and theories from the field. End with a line: "Answer: <N>", where N is an integer from 1 to 4.

**Legal:**

```
# Legal Domain Context
#
Given the legal context and questions provided, formulate a response that
- Specifically answers the question using relevant legal principles and concepts
- Avoids unnecessary assumptions or making scope autonomy assumptions
- Poses clear chain of argumentation

End with a line: "Answer: <N>", where N is an integer from 1 to 4.
```

### D.4.2   Without Chain-of-Thought

**Mathematics:**

```
For each given problem in this domain, breaking down the steps involved to
reach the final answer. Ensure calculations are correct and accurate. Consider
any constraints or limitations provided, and make assumptions as needed. DO NOT
show any reasoning and calculation in the output. ONLY OUTPUT THE FINAL
NUMERICAL ANSWER. End with a line: "Answer: <N>", where N is a number.
```

**Psychology:**

```
For each question in the provided list, please answer with a clear and concise
statement that draws on relevant concepts, theories, and methodological
principles from the domain of psychology, education, and research methods.

Assume a knowledge domain that includes:

* Aging and sexual functioning
* Piaget's theory on child development
* Standard error of measurement
* Teratogens and fetal development
* Training program development and evaluation
* Gestalt psychotherapy and client understanding
* Memory decline in aging
* Ethical considerations for mental health professionals
* Structural family therapy and boundary clarification
* Longitudinal research design and parent alcoholism

Fundamental mathematical and scientific principles are assumed to be known.
End with a line: "Answer: <N>", where N is an integer from 1 to 4.
```

**Legal:**

```
# Legal Domain Context
#
# Tasks:

## Answer the question with a clear and concise rationale
## Use relevant legal concepts and principles to support your answer
## Avoid unnecessary assumptions or expanding on unrelated topics

End with a line: "Answer: <N>", where N is an integer from 1 to 4.
```


