# OpenReview forum: "``You are a brilliant mathematician'' Does Not Make LLMs Act Like One"
_Agents4Science/2025/Conference — Agents4Science_

### Official Review · Reviewer_Cw2t · 2025-10-03
**.**

**Clarity:** 3
**Significance:** 2
**Originality:** 2
**Overall:** 4
**Confidence:** 3

**Summary:**

This paper analyzes the effect of persona prompting on task performance. Paper is well crafted and experiments seem to be well executed. This paper provide interesting insights for general AI usage and safety.

My only main comment is regarding the novelty of this work. Authors cited work report very similar results and thus I am not sure what's the contribution. Specifically, Zheng et al. (2023) already established that personas don't improve performance and effects are unpredictable. A better framing of the related work would help in assessing what's the value of this work that while interesting, does seem a little bit incremental. However, in defense of the paper, I could say that the experiment are done on modern models and thus it is interesting to see that personas do not work on modern models. There are some additional interesting parts to this paper (cross-domain experiments, model generated persona, negative personas) which are interesting and valuable.

My suggestion would be to lower the general claiming a bit. Specifically, the abstract and introduction should clarify these build on Zheng's empirical findings by explaining mechanisms (why personas fail through surface cues, not expertise) rather than discovering the problem. Focus on the other experiments that are indeed interesting theoretical contributions.

**Questions:**

.

**Ethical Concerns:**

.

**Limitations:**

.

**Quality:**

3

**Strengths And Weaknesses:**

.

---

### Official Review · Reviewer_AIRev1 · 2025-10-06
**AIRev 1**

**Confidence:** 5
**Overall:** 4
**Clarity:** 0
**Significance:** 0
**Originality:** 0

**Summary:**

Summary by AIRev 1

**Questions:**

N/A

**Ai Review Score:**

4

**Quality:**

0

**Strengths And Weaknesses:**

This paper presents a systematic, multi-domain evaluation of persona prompting versus domain priming across four LLMs and two reasoning modes. The study is broad and systematic, with multi-model, multi-domain, and CoT vs. no-CoT comparisons, and includes novel aspects such as negated personas and cross-domain persona transfer. The results are clearly visualized, and the paper provides actionable recommendations favoring domain priming over persona role-play for reliability. Reproducibility is supported by detailed prompt examples and appendices, and the work is well-situated within related literature.

However, a key experimental confound is present: the priming prompts contain more detailed procedural guidance and are longer than the persona prompts, making it difficult to attribute observed differences solely to persona vs. priming. The lack of length- and content-matched controls weakens the central claim. Statistical treatment is limited, with no confidence intervals or significance tests, and the datasets are described only at a high level. Baseline coverage and ablations are incomplete, and the risk of overgeneralization is noted. Code and data are not yet released, limiting reproducibility.

The paper is generally clear and well organized, though some claims overstate generality. Ethical considerations are discussed, but the use of real experts' names in prompts could be addressed further. The negation and cross-domain experiments are valuable, and the consistent priming advantage is a useful contribution, but stronger controls and statistics are needed to solidify the claims.

Actionable suggestions include controlling for prompt content and length, improving statistical rigor, expanding datasets and baselines, conducting mechanistic analyses, enhancing reproducibility, and tempering conclusions. Overall, this is a timely and relevant empirical study with thoughtful experiments and clear guidance, but substantial revisions are needed to address confounds and strengthen the evidence. The recommendation is borderline accept, contingent on addressing these issues.

---

### Official Review · Reviewer_AIRev2 · 2025-10-06
**AIRev 2**

**Confidence:** 5
**Overall:** 6
**Clarity:** 0
**Significance:** 0
**Originality:** 0

**Summary:**

Summary by AIRev 2

**Questions:**

N/A

**Ai Review Score:**

6

**Quality:**

0

**Strengths And Weaknesses:**

This paper presents a rigorous and comprehensive empirical investigation into the effectiveness of persona prompting for instructing large language models (LLMs). The authors systematically compare persona prompts to domain priming across mathematics, psychology, and law, using four state-of-the-art LLMs and two reasoning modes. The central finding is that persona prompting is volatile and often degrades performance, while domain priming provides modest but stable improvements. The experiments, including negated personas and cross-domain applications, suggest persona effects stem from surface-level linguistic artifacts rather than genuine expertise activation. The paper is significant for its practical recommendations, methodological rigor, and clear evidence, advocating for domain priming over persona prompting. Minor weaknesses include the lack of formal statistical significance tests, limited domain scope, and a missed opportunity for deeper analysis of model-generated prompts. Overall, this is a high-quality, impactful paper that is clearly recommended for acceptance.

---

### Official Review · Reviewer_AIRev3 · 2025-10-06
**AIRev 3**

**Confidence:** 5
**Overall:** 5
**Clarity:** 0
**Significance:** 0
**Originality:** 0

**Summary:**

Summary by AIRev 3

**Questions:**

N/A

**Ai Review Score:**

5

**Quality:**

0

**Strengths And Weaknesses:**

This paper presents a systematic evaluation of persona prompting across mathematics, psychology, and law domains using four state-of-the-art language models. The authors investigate whether instructing models to adopt expert roles (e.g., "you are a mathematician") actually improves performance compared to domain priming (non-persona cues) and baseline approaches.

Quality: The paper is technically sound with a well-designed experimental setup. The evaluation across four models (Gemini 2.5 Flash, GPT-4.1, Qwen3 32B, Llama 3.1 8B) and three domains provides good coverage. The inclusion of negated personas (e.g., "you are NOT a mathematician") is particularly clever and reveals important insights about the instability of persona effects. The cross-domain experiments effectively challenge the expertise activation theory. The model-generated optimization experiments add valuable depth by testing whether poor human design explains persona limitations.

Clarity: The paper is well-written and organized. The experimental methodology is clearly described, and the results are presented with appropriate visualizations. The extensive appendix with model-generated prompts enhances reproducibility. The writing effectively communicates the counterintuitive findings.

Significance: This work addresses a practically important question about a widely-used prompting technique. The findings have direct implications for AI deployment and prompt engineering practices. The demonstration that domain priming consistently outperforms persona prompting across models and domains is valuable for the community. The safety implications regarding deployment brittleness are well-articulated and important.

Originality: The systematic comparison of persona prompting vs. domain priming is novel, as is the use of negated personas to test robustness. The cross-domain experiments and model-generated optimization provide fresh insights. While persona prompting has been studied before, this comprehensive evaluation across multiple dimensions is original.

Reproducibility: The paper provides sufficient experimental details, including datasets, model settings, and prompt examples. The authors commit to releasing code and data. The extensive appendix with all prompts supports reproducibility.

Strengths:
- Comprehensive evaluation across multiple models and domains
- Novel use of negated personas exposes fundamental instability
- Cross-domain experiments effectively challenge expertise activation theory
- Practical implications are clearly articulated
- Strong experimental design with appropriate controls

Weaknesses:
- Limited to three domains - broader evaluation would strengthen claims
- Focus on accuracy metrics only - other aspects like response quality not assessed
- Some statistical analysis could be strengthened (though large sample sizes help)
- The temperature=0 setting may not reflect all use cases

Minor Issues:
- Some figures could benefit from error bars or confidence intervals
- The related work section could better position the work relative to recent persona research

The paper makes a valuable contribution by systematically debunking assumptions about persona prompting effectiveness and providing practical alternatives. The evidence for domain priming as a more reliable approach is compelling and will likely influence prompt engineering practices.

---

### Note · Reviewer_AIRevCorrectness · 2025-10-06

**Correctness Check**

### Key Issues Identified:

- Prompt-content confound: Priming prompts include substantial procedural guidance and formatting control, while persona prompts are short role cues. Without controlling instruction length and structure, it is unclear whether observed gains are due to domain priming or added scaffolding.
- Lack of statistical reporting: No confidence intervals or significance tests; legal set is small (~117 items), making small percentage differences hard to interpret. Checklist states Yes for statistical reporting despite admitting none were provided (page 21, Q7).
- Output-format inconsistency: Claimed standardized ‘Answer: <N>’ format (Section 2.1) is violated in at least one optimized priming prompt (Gemini CoT math exemplar uses ‘Answer: $11’, Appendix D, page 12) and final instruction omits the ‘Answer: <N>’ requirement—risking parsing errors and unfair comparisons.
- Insufficient evaluation harness details: The paper does not specify how noncompliant outputs were parsed, how currency/units were handled, or how failures were treated. This can differentially impact conditions with longer/structured prompts.
- Ambiguity in persona aggregation: Results show a single ‘Persona’ bar, but three persona types (generic, historical, modern) were introduced. The aggregation method (average, best-of, or a single type) is not described, limiting reproducibility and interpretability.
- Dataset provenance and splits: ‘~1,300 GSM8K-style’ and ‘~612 MMLU-Psychology’ items are not precisely specified; train/test splits, contamination checks, and exact selection criteria are missing. Closed-model API versioning/timestamps are not provided.
- Aggregation weighting: Figure 2’s ‘overall’ comparisons do not state whether macro- or micro-averaging was used across domains with large size disparities, which can bias totals.
- Model-optimized prompts: The 10 representative examples used to generate optimal prompts are not guaranteed to be disjoint from evaluation data; lack of clarity raises potential leakage/overfitting concerns.

---

### Note · Reviewer_AIRevRelatedWork · 2025-10-06

**Related Work Check**

No hallucinated references detected.

---

### Decision · Program_Chairs · 2025-10-08

**Decision:**

Accept

**Comment:**

Thank you for submitting to Agents4Science 2025! Congratualations on the acceptance! Please see the reviews below for feedback.